# Cheek-Pro-Heart: What Can the Buccal Mucosa Do for Arrhythmogenic Cardiomyopathy?

**DOI:** 10.3390/biomedicines11041207

**Published:** 2023-04-18

**Authors:** Carlos Bueno-Beti, Angeliki Asimaki

**Affiliations:** Molecular and Clinical Sciences Research Institute, St George’s, University of London, London SW17 0RE, UK; cbuenobe@sgul.ac.uk

**Keywords:** sudden cardiac death, arrhythmogenic cardiomyopathy, buccal mucosa, desmosomes, phospholamban, plakoglobin, connexin43

## Abstract

Arrhythmogenic cardiomyopathy (ACM) is a heart muscle disease associated with ventricular arrhythmias and a high risk of sudden cardiac death (SCD). Although the disease was described over 40 years ago, its diagnosis is still difficult. Several studies have identified a set of five proteins (plakoglobin, Cx43, Nav1.5, SAP97 and GSK3β), which are consistently re-distributed in myocardial samples from ACM patients. Not all protein shifts are specific to ACM, but their combination has provided us with a molecular signature for the disease, which has greatly aided post-mortem diagnosis of SCD victims. The use of this signature, however, was heretofore restricted in living patients, as the analysis requires a heart sample. Recent studies have shown that buccal cells behave similarly to the heart in terms of protein re-localization. Protein shifts are associated with disease onset, deterioration and favorable response to anti-arrhythmic therapy. Accordingly, buccal cells can be used as a surrogate for the myocardium to aid diagnosis, risk stratification and even monitor response to pharmaceutical interventions. Buccal cells can also be kept in culture, hence providing an ex vivo model from the patient, which can offer insights into the mechanisms of disease pathogenesis, including drug response. This review summarizes how the cheek can aid the heart in the battle against ACM.

## 1. Introduction

Arrhythmogenic cardiomyopathy (ACM) is a heart muscle disease associated with ventricular arrhythmias and sudden cardiac death (SCD). Pathologically, it is characterized by the gradual degeneration of cardiac myocytes (CMs) and their subsequent replacement by fat and fibrous tissue [1]. Initial reports revealed that the disease was caused by malformation of the right ventricular myocardium, and hence, it was first given the name arrhythmogenic right ventricular dysplasia (ARVD). Soon, it was realized that the right ventricular myocardium was correctly formed, and its degeneration started later on in life. The disease was thus classified as a cardiomyopathy and given the name arrhythmogenic right ventricular cardiomyopathy (ARVC). A few years later, it was recognized that the same process of myocyte degeneration and fibrofatty replacement can affect the right ventricle in isolation, the left ventricle in isolation or, in some cases, both ventricles. As a result, the disease is now known as arrhythmogenic cardiomyopathy (ACM) [2].

From an epidemiological standpoint, the prevalence of ACM ranges from 1:1000 to 1:5000 in the general population [3]. The large range is mainly attributed to difficulties in diagnosis. ACM shows a wide phenotypic variation with some individuals dying suddenly at an early age, others developing severe heart failure, while some may be carrying disease-causing variants and yet not manifest any clinical evidence of disease throughout their lifetime [4]. ACM, however, does appear to be more prevalent in specific regions of the world, perhaps owing to founder effects. One example is the Veneto region of Italy, where ACM is the top cause of SCD in the young, particularly in those under 35 years of age who are involved in endurance exercise [5]. At the moment, there is no gold standard for the diagnosis of the disease. An International Task Force set a list of criteria in 1994, which are divided into major and minor. The categories include pathological, imaging, ECG, genetic findings, as well as family history of disease. In order for a diagnosis to be made, a patient needs to fulfill criteria from different categories, which may be relatively specific but are not highly sensitive [6].

ACM is a very arrhythmogenic disease. Arrhythmias arise first, before the myocardium becomes structurally remodeled [7]. We refer to this early phase as ‘concealed’. This is a unique stage among non-ischemic cardiomyopathies. Patients diagnosed with hypertrophic cardiomyopathy (HCM) are at risk of arrhythmias because of cardiomyocyte disarray, fibrosis and small vessel disease. Patients diagnosed with dilated cardiomyopathy (DCM) are at risk of arrhythmias following significant remodeling and dysfunction of the ventricles. In contrast, patients diagnosed with ACM are at a high risk of potentially fatal ventricular arrhythmias even while their heart is still apparently morphologically normal. This is highly similar to what occurs in patients with cardiac channelopathies, hence increasing the complexity of the ACM spectrum [2,4]. In this review, we shall cover the protein biomarkers implicated in the pathogenesis of ACM to date and how the analysis of their distribution in the buccal mucosa can aid the diagnosis and understanding of the disease.

## 2. Genetics of ACM

The major step forward in ACM genetics took place in the year 2000. Naxos disease is a syndromic form of ACM associated with skin and hair abnormalities. It shows 100% penetrance and hence constituted an ideal model for early linkage analysis. Consequently, the first ACM-causing gene was identified as plakoglobin (*JUP*). *JUP* codes for a desmosomal protein (PG). Patients with Naxos disease are homozygous for a C-terminal deletion in the gene [8,9]. Carvajal syndrome is another syndromic form of ACM. Specifically, these patients show a form of DCM, as well as skin and hair abnormalities. Soon after the identification of *JUP* as an ACM-causing gene, Carvajal syndrome was found to be underlied by a deletion in the desmoplakin gene (*DSP*), coding for another desmosomal protein [10]. Since these early discoveries, more mutations were identified in *JUP* and *DSP* in ACM patients showing dominantly inherited forms of the disease [11,12,13,14,15]. Desmosomes are sites of cell–cell coupling. In cardiac myocytes, they are found lying in the intercalated discs (IDs). These are the areas of contact between cardiac myocytes, which make sure the adjacent cells are strongly coupled (thus ensuring they can withstand physical stress) and are also responsible for electrical coupling, and thus, synchronous contraction. Such structures are mainly found in tissues, which are exposed to high levels of mechanical stress, primarily the skin and the cardiac muscle. There, they ensure adhesion by tethering the intermediate filaments of the cytoskeleton [16].

These early discoveries naturally focused the community’s attention on further components of the desmosome. Soon, numerous mutations were identified in the genes coding for the three additional major desmosomal proteins: plakophilin-2 (*PKP2*), desmoglein-2 (*Dsg2*) and desmocollin-2 (*Dsc2*) [17,18,19]. At the moment, more than 60% of patients diagnosed with ACM worldwide are shown to carry variants in the aforementioned genes. Accordingly, ACM is considered to be a ‘disease of the desmosome’ [20].

Non-desmosomal genes have also been implicated in the pathogenesis of ACM. Adherens junctions is another class of intercellular adhesion complexes, which mechanically couple CMs [16]. Mutations in the *CDH2* and *CTNNA3* genes (coding for the adherens junctions proteins N-cadherin and catenin-α3, respectively) have been identified in a small number of ACM patients [21,22,23,24]. Mutations in genes encoding for cytoskeletal components have also been described, specifically in desmin (*DES*), filamin C (*FLNC*) and titin (*TTN*) [25,26,27]. Interestingly, a large number of ACM patients originating from Newfoundland, Canada, are bearing a mutation in transmembrane protein 43 (*TMEM43*; S358L), an inner nuclear membrane protein shown to interact with emerin and lamins. *TMEM43* mutation carriers exhibit a highly aggressive form of the disease characterized by high SCD incidence, male predominance and LV involvement [28]. The phospholamban (*PLN*) mutation R14del, first described in Greek families with DCM and heart failure [29], was identified in 2012 in a large number of Danish patients with a clinical diagnosis of ACM or DCM, supporting a broader phenotypic spectrum for ACM [30]. PLN is a regulator of the sarcoplasmic reticulum Ca^+2^ ATPase 2a (SERCA2a) pump in the myocardium and thus crucial for maintaining Ca^+2^ homeostasis [30]. Finally, rare, isolated ACM-causing mutations have been identified in the *SCN5A* and *TGF3β* genes. *SCN5A* codes for the major protein component (Nav1.5) of the cardiac sodium channels, while *TGF3β* encodes transforming growth factor 3β, a cytokine involved in the regulation of cell adhesion and formation of the extracellular matrix. The frequency of these genetic alterations in large patient cohorts is, however, very low [31,32].

Typically, ACM has a relatively late onset (late adolescence/early adulthood). However, multiple mutations (digenic or compound heterozygosity) have been associated with earlier onset, more aggressive disease and a higher incidence of SCD [33].

## 3. Proteins ‘Shifting’ in the Heart

The vast majority of the world’s myocardial tissue bank consists of heart samples, which were obtained either during autopsies or through endomyocardial biopsy (EMB) procedures. Such samples are typically fixed in formalin and maintained in paraffin cassettes. This method of preservation limits the range of experimental techniques that can be used. Nevertheless, analyzing this patient material via immunohistochemistry has greatly improved our understanding of ACM pathogenesis by uncovering key protein biomarkers [5].

The first molecular studies were performed on heart samples from patients with a diagnosis of Naxos disease. These studies showed that the mutant protein, PG, was expressed at normal levels. However, it was not distributed at the IDs [34]. Next, immunohistochemical studies were performed on a heart sample from a patient diagnosed with Carvajal syndrome. Once again, the mutant protein, in this case DSP, was unable to reach the IDs [35]; interestingly though, neither was PG, although it was not itself genetically altered [35]. In agreement with this initial insight, the immunoreactive signal for PG was subsequently shown to be significantly reduced at cell–cell junctions in heart samples from a large number of ACM patients, independent of which gene was driving the disease phenotype [36].

Those first immunohistochemical studies also examined the distribution of connexin43 (Cx43). Coded by the *GJA1* gene, Cx43 is the major protein subunit of the electrical gap junctions in the ventricles. Interestingly, the immunoreactive signal for the gap junction protein was significantly reduced at the cardiac cell–cell junctions, indicating failure of Cx43 to fully incorporate at the IDs as well [34,35]. Since these studies, more reports have shown reduced Cx43 signal, known as gap junction remodeling, in heart samples from patients with autosomal dominant forms of ACM. This finding is in agreement with the belief that the formation of gap junctions is dependent on the correct formation and maintenance of mechanical junctions, such as desmosomes and adherens junctions [36,37]. This is not the first time that a heart muscle disease is shown to be characterized by gap junction remodeling; indeed, it has also been shown in HCM, DCM and ischemia [38,39]. In these diseases though, gap junctions are remodeled after the heart has been structurally altered. On the contrary, in ACM, gap junctions are remodeled before the onset of histological and major structural changes. This difference suggests that gap junction remodeling may have a fundamental role in underlying the arrhythmogenic substrate of early ACM stages [40].

The immunoreactive signal for PG was also shown to be significantly depressed at the IDs in myocardial samples from three ACM patients bearing the S358L mutation in the *TMEM43* gene [41]. The signal for Cx43, PKP2 and emerin appeared strong at the IDs and indistinguishable from controls [41]. Although the implications of these findings are still uncertain, they do seem to suggest that redistribution of PG from junctional to intracellular pools is a fundamental, consistent feature of the ‘final common pathway’ in ACM pathogenesis [5].

As mentioned in the previous section, the R14del mutation in the *PLN* gene underlies both cases of ACM and DCM. Interestingly, the ID-localized immunoreactive signal for PG is depressed only in myocardial samples from patients with a diagnosis of ACM but not with DCM [30]. This observation suggests that PG redistribution from junctional to intracellular pools follows the patient’s phenotype and not their genotype. In agreement with this, PG distribution was not affected in myocardial samples from a majority of patients bearing *FLNC* variants associated with a phenotype of arrhythmogenic DCM as opposed to classical ACM [42,43]. Reduced immunoreactive signal for PG and Cx43 has since been reported in further ACM patient series [44,45,46,47].

An additional ACM protein biomarker was introduced by Noorman et al. in 2013, who showed reduced ID localization for Nav1.5 in the hearts of ACM patients. Similar to Cx43, decreased levels of Nav1.5 may contribute to arrhythmia vulnerability, particularly in the early, concealed form of the disease [48]. In agreement with these findings, decreased sodium current has been recorded in a range of experimental models for ACM, including patient-derived induced pluripotent stem cell (iPSC)-CMs [49,50,51,52].

Nav1.5 is targeted to the cell surface by a transporter known as PDZ domain-containing protein synapse-associated protein-97 (SAP97, *DLG1*) [53]. In the myocardium of healthy controls, the immunoreactive signal for SAP97 is concentrated at the IDs and also shows a sarcomeric pattern. By contrast, SAP97 signal is markedly reduced in myocardial samples from ACM patients, independent of the disease-causing mutation [51]. SAP97 appeared to retain its sarcomeric localization (although there was a modest decrease in the ID signal intensity) in myocardial samples from patients with HCM, DCM or ischemic cardiomyopathy. Accordingly, SAP97 signal appears to be reduced primarily in ACM hearts compared to other forms of myocardial disease, establishing it as the fourth ACM protein biomarker [51].

The zebrafish constitutes an ideal model for high-throughput screening of small molecules. The results obtained can then be readily tested in cellular or mouse disease models. A zebrafish model expressing the *JUP* mutation underlying Naxos disease was used in such a screen. A small molecule inhibitor of glycogen synthase kinase 3β (GSK3β), SB216763, was shown to prevent and reverse the disease phenotype in the mutant fish. Subsequently, SB216763 was shown to prevent quantifiable disease end points in further models of ACM, including cell culture models, mouse models and iPSC-derived cardiac myocytes. Unfortunately, the extrapolation of these findings to patients is limited, as suppression of GSK3β can cause cancer [51,54]. Even if this discovery did not lead to a mechanism-based therapy, it did show that GSK3β plays a pivotal role in the pathogenesis of ACM, and it did provide us with a fifth biomarker [54]. In the myocardium from healthy controls, GSK3β resides diffusely in the cytoplasm. On the contrary, in the myocardium from patients with a diagnosis of ACM, GSK3β is concentrated at the IDs. This was found in samples from a large number of ACM patients and was consistent regardless of the disease-causing mutation. It also appeared to be specific to this disease, since GSK3β retained its cytoplasmic localization in samples from patients with other forms of heart disease [54].

Not all of the aforementioned protein transpositions are highly specific for ACM. Their combination, however, does provide us with an ACM molecular signature. Classical ACM is characterized by the inability of plakoglobin, connexin43, Nav1.5 and SAP97 to fully incorporate in the IDs and by a shift of GSK3β distribution from the cytoplasm to the cell–cell junctions. Still, the value of this signature in living patients was heretofore restricted, since such an analysis would require a piece of one’s heart. EMB sampling is risky and invasive, shows a low diagnostic yield and is currently used as a method of ‘last resort’ [55]. To bypass this limitation, a surrogate tissue was required. As mentioned above, desmosomes are particularly prominent in those tissues, which are exposed to mechanical forces, such as the skin and the heart [16]. The junctional signal for PG was shown to be depressed in skin samples from children with severe ACM manifestation, but patients are not routinely subjected to skin biopsy sampling unless they also show cutaneous abnormalities [56,57]. The buccal epithelium, however, is a tissue, which can be accessed very easily and can be sampled in a minimally invasive and risk-free manner [58].

## 4. The Buccal Mucosa in Adult ACM

The buccal mucosa consists of flattened, squamous cells with small central nuclei and clearly identifiable edges. In normal subjects, these cells show strong membrane immunofluorescent signal for PG, DSP, Cx43 and plakophilin-1 (*PKP1,*
Figure 1). PKP1 is one of the three plakophilin isoforms. It is primarily expressed in the upper epithelia. Conversely, PKP3 is primarily found in the lower epithelia, while PKP2 is the isoform expressed in the heart [59].

To determine whether the signal for these proteins is redistributed in the buccal mucosa in ACM, smears were obtained from a large number of patients bearing desmosomal gene mutations and subjected to immunocytochemistry. The membrane signal for PG was significantly depressed in 34/39 ACM subjects, while the membrane signal for Cx43 was virtually absent in 30/31 cases available. All proteins examined showed normal distribution in smears obtained from 40 control subjects. These were defined as individuals with no clinical signs or family history indicative of heart disease [59]. The signal for PKP1 was depressed in all patients bearing mutations in *PKP2* but no patients with mutations in *DSP*, *Dsc2*, *JUP* or *Dsg2*. Conversely, the immunoreactive signal for DSP was reduced in all patients bearing mutations in *Dsc2*, *Dsg2* or *DSP* but no patients with mutations in *PKP2* or *JUP*.

As part of this study, 15 silent carriers were also sampled. These were individuals shown to carry a disease-causing mutation but without any clinical evidence of ACM. The signal for PG was depressed in 12/15 and the signal for Cx43 in 14/15 such carriers. Finally, the signal for PG was normal in smears from six patients diagnosed with different heart diseases (HCM, DCM and ischemic cardiomyopathy), while the signal for Cx43 was depressed in all six sample sets [59]. Collectively, this study suggested a close relationship between the patterns of altered distribution of ID proteins in CMs and cell–cell junction proteins in buccal mucosa cells in patients with ACM. Interestingly, the study also showed that the localization of key proteins in buccal cells is correlated with the gene, which is driving the disease. Interestingly, PKP1-3 are isoforms of the same protein but are encoded by separate genes. In fact, those genes even reside on separate chromosomes. The fact that mutations in the PKP2-coding gene (which is only expressed in the heart) can alter the distribution of PKP1 (which is only found in the upper epithelia) suggests that the two genes are governed by yet unknown common regulatory pathways.

In a subsequent study, Driessen et al. obtained buccal smears from two patient groups in the Netherlands; the first group was diagnosed with ACM and found to carry desmosomal gene variants, while the second group was diagnosed with ACM associated with the R14del variant in *PLN*. In the first group, the PG signal was significantly depressed in the buccal mucosa in both symptomatic ACM patients and asymptomatic mutation carriers compared to sex-matched controls (*p* = 0.002) [60]. When the amount of PG labeling was scored, a significant correlation was revealed between signal reduction in the buccal mucosa and the 2010 Task Force Criteria scores [6]; the most severely affected ACM patients showed the largest reduction in PG. Moreover, lower PG scores strongly correlated with the number of premature ventricular contractions on 24 h Holter monitoring (*p* = 0.02). However, these correlations were apparent only at a specific antibody dilution, highlighting the need for adherence to a specific research protocol for accurate data generation [60].

Plakoglobin distribution appeared normal and indistinguishable from controls in seven PLN-R14del carriers with a clinical diagnosis of DCM and in seven asymptomatic carriers of the mutation. The signal for PG was, however, significantly depressed in 3/3 R14del carriers with a clinical manifestation of ACM [60]. This is in agreement with the observations made in heart samples from R14del carriers, where a clinical diagnosis of ACM correlated with PG remodeling while DCM did not [30].

## 5. The Buccal Mucosa in Pediatric ACM

Up until recently, it had not been possible to correlate molecular changes, including redistribution of selected proteins, with the onset of disease in previously silent carriers or with disease progression/deterioration over time. It had also not been possible to assess the effect of medication on similar molecular processes. As mentioned above, EMB procedures are difficult, risky and rarely performed on patients. Additionally, serial EMBs on a patient are even rarer. Obtaining a heart sample, however, from an individual who is a mutation carrier only, without showing the actual disease, is ethically unthinkable. This becomes even more of a science fiction scenario when the silent carriers are children. The use of buccal cells as a mirror of heart cells, however, has provided the opportunity to address these questions [61].

Key junctional proteins alter their distribution, shifting from the cell borders to the cell interior in buccal cells of children with a diagnosis of desmosomal ACM, as shown in adults [61]. Representative pictures are shown in Figure 2 [61].

Interestingly, however, the localization of all examined proteins was normal and indistinguishable from controls in children carrying desmosomal gene variants without clinical manifestation of disease. The ease of buccal smear preparation allows for serial sampling, and hence, conduction of much needed molecular longitudinal studies. In the same study, Bueno-Beti et al. showed that, in addition to the initial protein set shifting at disease onset, further key protein distribution changes occur with clinical deterioration. Finally, the authors describe a case of an ACM patient initially showing reduced membrane localization of Cx43 [61]. An increased arrhythmic load prompted the initiation of a beta blocker. Holter monitoring several months later showed significant reduction in arrhythmias. A second set of buccal smears was obtained, and the localization of Cx43 now appeared membranous and indistinguishable from controls [61]. Representative images from this patient are shown in Figure 3. In summary, this study showed that analysis of buccal cells may be a straightforward, non-invasive and cost-effective way to confirm a diagnosis of ACM, to mark the clinical onset of disease, follow its progression over time and even assess whether therapeutic interventions show efficacy [61].

## 6. Cheeks beyond Diagnosis

The studies described herein suggest that analysis of key protein distribution in buccal cells may be used as a diagnostic and prognostic tool. It may also aid genetic predictions. Buccal cells, however, can also serve as a patient-derived platform for personalized medicine. Buccal cells are terminally differentiated, hence incapable of dividing. They can, however, be maintained in culture for up to 10 days. Asimaki et al. collected buccal cells from six patients with a clinical diagnosis of ACM and two silent carriers of desmosomal gene mutations, yet to manifest a disease phenotype. The cells were cultured and immunostained for key junctional proteins. The signal for PG and Cx43 was significantly reduced at the cell membranes compared to controls [59]. In the past, it had been shown that when neonatal rat ventricular myocytes transfected to express ACM-causing desmosomal gene mutations are exposed to SB216763 for 24 h, the distribution of key junctional proteins is restored [51]. Similarly, cultured buccal cells from patients and carriers were exposed to the GSK3β inhibitor for 24 h. The distribution of PG and Cx43 was restored to control patterns [59]. This suggests that, in terms of junctional protein distribution, it is the same pathways that govern both the heart and the cheek’s interior. The fact that patient-derived buccal cells can be maintained ex vivo provides a platform to investigate mechanistic pathways and perform personalized drug screens [59].

## 7. Conclusions

Although it has now been over 40 years since ACM was first described, establishing an accurate diagnosis and performing reliable risk stratification remain highly challenging actions. It has been shown that a set of proteins are consistently re-localized in the hearts of patients with ACM. This has aided post-mortem diagnosis of SCD victims. However, the use of this protein signature in living patients has been limited because the analysis requires a sample from one’s heart. Obtaining a heart sample is a highly risky and difficult procedure, only performed in highly specialized centers and only if deemed absolutely necessary. Interestingly, when it comes to the distribution of junctional proteins, buccal cells behave in the same way as heart cells. Accordingly, they can be used as a mirror of the heart and help diagnosis, risk stratification, even monitor how patients respond to pharmaceutical therapy. In addition to these, buccal cells can also be maintained in culture and thus offer an ex vivo model from the patient, which can be used to provide mechanistic insights and perform personalized drug screens. More research is required before buccal cells can be established as a diagnostic/prognostic tool or a patient-specific drug platform. The results summarized herein, however, suggest that the cheek can aid the heart in reducing the burden of SCD in our society.

## Figures and Tables

**Figure 1 biomedicines-11-01207-f001:**
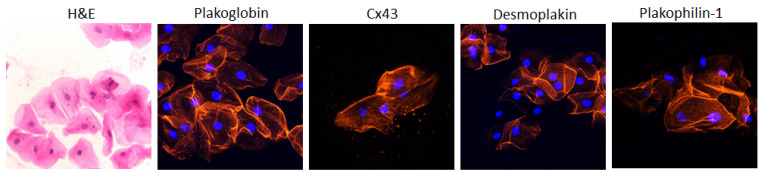
Representative confocal immunofluorescence images of buccal epithelial cells obtained from healthy controls. In the first image, cells are stained with hematoxylin and eosin (H&E). They show the typical squamous morphology; they have small central nuclei; and their edges are clearly identified. The remaining images show control buccal cells immunostained for plakoglobin, Cx43, desmoplakin and PKP1. The immunoreactive signal for all four proteins is strong and clearly localizes at the cell borders. Cell nuclei (blue) were stained with DAPI [59].

**Figure 2 biomedicines-11-01207-f002:**
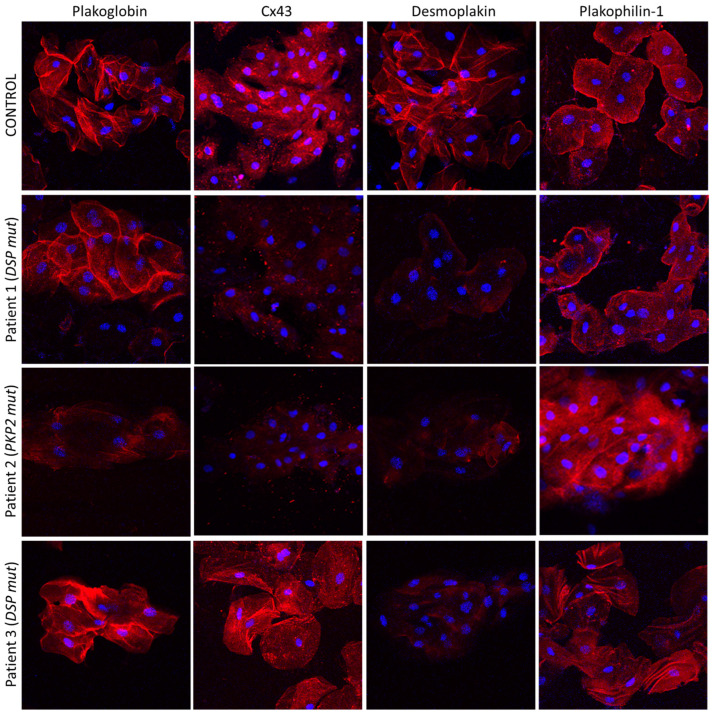
Confocal immunofluorescence images of buccal cells obtained from a healthy child (control) and three children with a clinical diagnosis of ACM. Patient 1 is bearing a pathogenic mutation in the *DSP* gene. He shows significantly reduced membrane signal for DSP and Cx43 compared to the control, but the distribution of PG and PKP1 appears to be normal. Patient 2 is carrying a pathogenic *PKP2* mutation. She shows decreased localization of PG, DSP and Cx43 at the membranes, but the localization of PKP1 is indistinguishable from the control. Finally, Patient 3 carries a disease-causing mutation in *DSP*. DSP does not localize at the cell borders but PG, PKP1 and Cx43 do. The nuclei are stained blue with DAPI [61].

**Figure 3 biomedicines-11-01207-f003:**
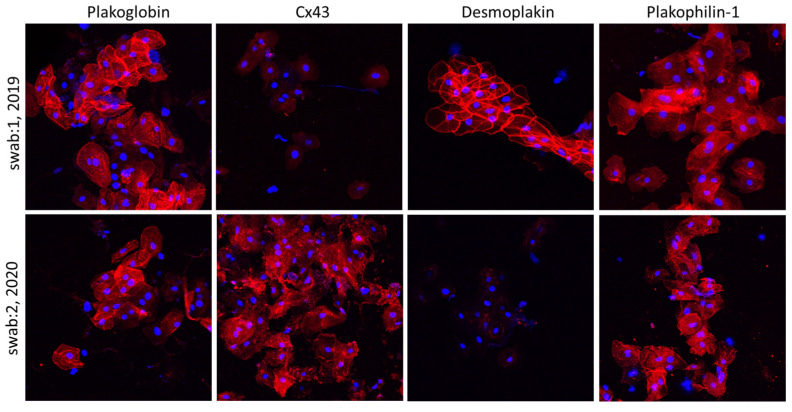
Confocal immunofluorescence images of two sets of buccal smears obtained from a child with a diagnosis of ACM bearing a *DSP* mutation. The first set of samples shows control localization for DSP, PG and PKP1, but membrane signal for Cx43 is virtually absent. In the second set of samples, obtained 18 months later, the immunoreactive signal for DSP appears to be missing from the membranes, while the distribution of Cx43 is now corrected to control-looking levels. Restoration of Cx43 signal correlated with a favorable response to beta blockers’ initiation [61].

## Data Availability

Not applicable.

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
