# Peer review of "Cheek-Pro-Heart: What Can the Buccal Mucosa Do for Arrhythmogenic Cardiomyopathy?"

_biomedicines, 2023, doi:10.3390/biomedicines11041207_

Round 1
Reviewer 1 Report
The subject of the article is very interesting. Starting from the ideea that Arrhythmogenic cardiomyopathy is a primary myocardial disease characterized clinically by ventricular arrhythmias and sudden cardiac death, in this review the authors are presenting the protein biomarkers implicated in the pathogenesis of Arrhythmogenic cardiomyopathy to date and how analysis of their distribution in the buccal mucosa can aid the diagnosis and understanding of the disease.
Previous studies have shown that buccal cells behave similarly to the heart in terms of protein relocalization. Clinical/practical implications of the study are related to the fact that protein shifts are associated with disease onset, deterioration and favorable response to anti-arrhythmic therapy. So, buccal cells can be used as a surrogate for the myocardium to aid diagnosis, risk stratification and even monitor response to pharmaceutical interventions.
The importance of the study is all the greater as it opens new therapeutic perspectives for this category of patients.
The article is very well structured, data are presented with scientific accuracy.
But, even into the abstract is mentioned something about the fact that ” buccal cells can be used as a surrogate for the myocardium to aid diagnosis, risk stratification and even monitor response to pharmaceutical interventions” the manuscript does not mention possible correlations with the clinical data, the evolution of the patients and pharmaceutical interventions.
The phrase: ”Collectively, these results suggest that analysis of buccal cells may be a safe, inexpensive, and non-invasive tool to confirm diagnosis, predict prognosis and potentially assess efficacy of pharmaceutical interventions in individuals carrying pathogenic variants linked to ACM” (between line no 252- line 254) is too general.
Author Response
Responses to Reviewer 1:
Many thanks to the reviewer for such a positive critique. Below we address each of the two points raised in the best of our abilities.
- The manuscript does not mention possible correlations with the clinical data, the evolution of the patients and pharmaceutical interventions.
We deeply thank the reviewer for this comment. More information has now been added to the main body of the manuscript showing correlation of junctional protein distribution with clinical status of the patients (specifically for reference 60; showing that when the amount of PG labelling was scored, a significant correlation was revealed between signal reduction in the buccal mucosa and Task Force Criteria scores. Moreover, lower PG scores strongly correlated with the number of premature ventricular contractions in 24hr Holter monitoring; lines 263-268). It is now also highlighted that ‘’ in addition to the initial protein set shifted at disease onset, further key protein distribution changes occur with clinical deterioration (lines 306-307), while a case is described showing restoration of Cx43 distribution in response to anti-arrhythmic therapy (lines 307-313).
- The phrase: ”Collectively, these results suggest that analysis of buccal cells may be a safe, inexpensive, and non-invasive tool to confirm diagnosis, predict prognosis and potentially assess efficacy of pharmaceutical interventions in individuals carrying pathogenic variants linked to ACM” (between line no 252- line 254) is too general.
We thank the reviewer for raising this point. This phrase has now been rewritten to: ‘’ In summary, this study showed that analysis of buccal cells may be a straightforward, non-invasive and cost-effective way to confirm a diagnosis of ACM, to mark the clinical onset of disease, follow its progression over time and even assess if therapeutic interventions show efficacy [61]. ‘’ (lines: 313-316).
Reviewer 2 Report
Congratulations!
I am pleased to learn something new. It is very well written with an excellent demonstration of Plakoglobin in the buccal mucosa.
Can you please describe the current practice to diagnose ARVC?
How the practice differed in Children vs. adults?
What various guidelines, such as from ESC or AHA/ACC, suggest the role of buccal mucosa study to diagnose ARC?
What is the modality in your practice? Because we still use genetic testing from blood.
Author Response
Responses to Reviewer 2:
Many thanks to the reviewer for such a positive critique! Below we address each of the questions in the best of our abilities.
- Can you please describe the current practice to diagnose ARVC?
We thank the reviewer for bringing up this important point! We have now added the following paragraph to address it: ‘’ At the moment there is no gold standard for the diagnosis of the disease. An International Task Force set a list of criteria in 1994, which are divided into major and minor. Categories include pathological, imaging, ECG, genetic findings as well as family history of disease. In order for a diagnosis to be made, a patient needs to fulfill criteria from different catego-ries, which may be relatively specific but are not highly sensitive [6].’’ (lines: 51-56)
- How the practice differed in Children vs. adults?
We thank the reviewer for this important question. The practice does not generally differ in children versus adults. The disease, however, is mostly manifest in late adolescence/early adulthood so we tend to see much fewer children with an ACM diagnosis in the clinics. Perhaps the only significant difference is that although T-wave inversions in precordial leads V1 and V2 are considered a minor diagnostic criterion in adults, their presence is normal on an ECG of a child under the age of 12. Since, however, the purpose of this review was to summarize protein biomarkers and their extrapolation from the heart to the cheek, we did not want to add much more on the current diagnostic practice. We hope the reviewer agrees with this decision.
- What various guidelines, such as from ESC or AHA/ACC, suggest the role of buccal mucosa study to diagnose ARVC?
We thank the reviewer for this question. The use of buccal cells as a diagnostic tool in ACM is a very recent finding that at the moment remains experimental. Although several groups around the world are now starting to perform similar studies, it is not yet at a stage to constitute part of ESC/AHA guidelines.
- What is the modality in your practice? Because we still use genetic testing from blood.
Many thanks to the reviewer for this important point of discussion. Analysis of buccal cells does not by any means replace genetic testing from blood. We, as well, continue to obtain blood samples and performing mutation analysis. In fact, it is only after the patient has been shown to have a variant in an ACM-causing gene, that we obtain a set of buccal smears to study the distribution of key proteins. In children that are still silent carriers, we obtain buccal smears twice a year to establish a correlation with onset of clinical disease while in adults with established disease, we obtain samples once a year, at their regular hospital appointment, and establish correlations between further protein shifts and clinical signs of disease deterioration (such as hot phases).